# RNA-Binding Proteins in Pulmonary Hypertension

**DOI:** 10.3390/ijms21113757

**Published:** 2020-05-26

**Authors:** Hui Zhang, R. Dale Brown, Kurt R. Stenmark, Cheng-Jun Hu

**Affiliations:** 1Cardiovascular Pulmonary Research Laboratories, Departments of Pediatrics and Medicine, School of Medicine, University of Colorado Anschutz Medical Campus, Aurora, CO 80045, USA; Hui.Zhang@cuanschutz.edu (H.Z.); dale.brown@cuanschutz.edu (R.D.B.); kurt.stenmark@cuanschutz.edu (K.R.S.); 2Department of Craniofacial Biology School of Dental Medicine, University of Colorado Anschutz Medical Campus, Aurora, CO 80045, USA

**Keywords:** pulmonary hypertension, RNA-binding proteins, alternative splicing, RNA metabolism, RNA therapeutics

## Abstract

Pulmonary hypertension (PH) is a life-threatening disease characterized by significant vascular remodeling and aberrant expression of genes involved in inflammation, apoptosis resistance, proliferation, and metabolism. Effective therapeutic strategies are limited, as mechanisms underlying PH pathophysiology, especially abnormal expression of genes, remain unclear. Most PH studies on gene expression have focused on gene transcription. However, post-transcriptional alterations have been shown to play a critical role in inflammation and metabolic changes in diseases such as cancer and systemic cardiovascular diseases. In these diseases, RNA-binding proteins (RBPs) have been recognized as important regulators of aberrant gene expression via post-transcriptional regulation; however, their role in PH is less clear. Identifying RBPs in PH is of great importance to better understand PH pathophysiology and to identify new targets for PH treatment. In this manuscript, we review the current knowledge on the role of dysregulated RBPs in abnormal mRNA gene expression as well as aberrant non-coding RNA processing and expression (e.g., miRNAs) in PH.

## 1. Introduction

Pulmonary hypertension (PH) is a progressive disorder that can ultimately lead to right heart failure and death. PH is defined by hemodynamic alterations of the pulmonary circulation, in which the mean pulmonary artery pressure (mPAP) is in excess of 20 mmHg [1]. Pulmonary vascular remodeling is a characteristic of chronic PH and is both the result of and a contributor to increased pulmonary vascular pressure through its effects on pulmonary vascular resistance (PVR). In this process, pulmonary arteries, both large and small, undergo significant structural alterations including intimal fibrosis, medial hypertrophy, and adventitial/perivascular inflammation and fibrosis [2]. The prevalence of PH is increasing in both the pediatric and adult population and it remains a significant cause of morbidity and mortality, which represents a substantial and growing health care burden [3,4,5]. In addition, PH is also a prevalent co-morbid condition that significantly worsens morbidity and mortality in patients with a variety of disorders, from acute lung injury to pulmonary fibrosis and chronic obstructive pulmonary disease (COPD). Unfortunately, effective therapeutic strategies are limited, as mechanisms underlying PH pathophysiology, especially aberrant gene expression in pulmonary vascular cells, remain unclear [6].

Cellular identity, morphology, and phenotype are governed by precise patterns of gene expression. The abnormal gene expression pattern in PH vascular cells is responsible for pulmonary vascular remodeling. Aberrant expression and/or activity of gene expression regulators in PH vascular cells are key drivers for PH initiation and progression [6,7]. Gene regulation is primarily carried out by proteins that bind DNA or RNA molecules at specific nucleotide sequences or structures. One class of such proteins are transcription factors (TFs), which bind short DNA sequences to regulate transcription in response to genetic, epigenetic, and environmental factors. We have previously reviewed the role of TFs in PH development and maintenance [7]. Another class is RNA-binding proteins (RBPs) that use their RNA-binding domain to recognize and bind to their target RNAs (double or single-stranded RNA) [8]. RBPs can regulate gene transcription but primarily act on RNA processing, including mRNA splicing, polyadenylation, translocation, stability, translation, as well as biogenesis of non-coding RNAs [9,10,11]. Differing from TF-mediated transcription regulation, which often takes a long time, RBP-mediated post-transcriptional regulation allows for relatively rapid response to demands in protein change [12,13].

Due to their critical roles in gene transcription and post-transcriptional regulation, RBPs are critical in human biology and development. Functional disruption of RBPs thus impacts on multiple human diseases including genetic disease [14], immune-metabolic diseases [15], cancer [8], and neurological diseases [16,17,18]. Recently, de Bruin and colleagues summarized the emerging roles of RBPs in systematic cardiovascular disease [19]. They reported that RBPs, including quaking (QKI), human antigen R (HuR), muscleblind, and serine/arginine-rich splicing factor 1 (SRSF1), play versatile roles in regulating the transcriptome in coronary artery vascular cells, inflammatory cells, and cardiomyocytes, which are important in cardiovascular health, adaptation, and disease. However, the biological significance of RBPs in pulmonary vascular disease remains largely unknown. Emerging evidence suggests that some RBPs are dysregulated, which influence the development and progression of PH. Herein, we will briefly review the current knowledge on the role of RBPs in PH and discuss the potential of therapies targeting RNA metabolism for the treatment of these life-threatening disorders.

## 2. RBPS in RNA Metabolism

RNA metabolism is a broad term that encompasses all aspects of the RNA life cycle, including synthesis, folding/unfolding, modifications, processing, and degradation. RBPs are involved in the regulation of each of these processes. In this section, we will summarize the general role of RBPs in RNA metabolism.

Although TFs and chromatin structure play an essential role in the regulation of gene transcription, evidence suggests that specific RBPs and their binding partners, the regulatory RNAs, particularly the long non-coding RNAs (lncRNAs), are also involved in transcriptional control and chromatin structure. Using RBP chromatin immunoprecipitation and DNA sequencing (ChIP-seq), a recent study demonstrates that active chromatin regions, particularly at gene promoters, are often loaded with RBPs. Further, the presence of RBPs in these regions usually indicates a high transcriptional output [20]. 

For protein-coding genes, the transcriptional product, the pre-mRNA transcript(s), are also extensively processed in a RBP-dependent manner [21]. Both ends of a pre-mRNA are modified by adding chemical groups, using protein complexes that bind to pre-mRNA. The group at the beginning (5’ end) is called a cap, while the group at the end (3’ end) is called a tail (polyadenylation). Both the cap and the tail make the transcript more stable and are also important for protein translation, and both groups are binding sites for RBPs (see below) [22]. 

Another major pre-mRNA processing event is RNA splicing, which is also heavily regulated by RBPs. The pre-mRNA splicing is carried out by RNA and protein complexes, called the spliceosome. The spliceosome consists of five small nuclear ribonucleoprotein particles (U1, U2, U4, U5, and U6 snRNPs). Using small nuclear RNAs as well as proteins within the snRNPs, the spliceosomes recognize and bind to the short consensus sequences surrounding the junctions between exon and intron, called 5′-(GU) and 3′-(AG) splice sites, and catalyze the removal of the intron and ligation of the exons. This process is called general or constitutive pre-mRNA splicing [23]. The process of general splicing can be regulated by numerous splicing auxiliary proteins that interact with RNAs and/or proteins within snRNPs to enhance or decrease snRNP’s association with 5′ and/or 3′ splice site(s) and/or to regulate conformational changes during spliceosome assembly and catalysis, often without binding to pre-mRNA. Due to the degenerate nature of the 5′ and 3′ splice sites in most eukaryotic pre-mRNAs, spliceosome recruitment and binding to the 5′ and 3′ splice sites of the pre-mRNA can be regulated by additional cis-acting RNA sequence elements and trans-acting RBPs [24,25,26]. The action of these other cis-acting RNA sequence elements (splicing enhancers and silencers, located in the intron and/or exon of the pre-mRNA) and trans-acting RBPs (splicing regulators: splicing activators bind to splicing enhancer while splicing repressors bind to splicing silencers) generate alternative splicing (AS). The best-known splicing regulators include members of the heterogeneous ribonucleoproteins (hnRNPs) and the serine/arginine-rich protein (SR protein) families. The hnRNPs bind to splicing silencers and promote exon skipping, and are called AS repressors. In contrast, the SR proteins bind to splicing enhancers and promote exon inclusion, and are called AS activators [27]. AS allows each gene to encode multiple mRNA variants that may code no proteins or proteins with different or even opposite activities. Thus, AS is one of the principal mechanisms for developmentally regulated and pathological cellular processes, and AS regulators such as hnRNPs and SR proteins are critically important in physiology and disease.

Once mRNA is generated in the nucleus, mRNA is translocated into the cytoplasm. mRNA export is achieved through the action of specific mRNA export receptors (also RBPs) that bind to RNA but also interact with the nuclear pore complex. Once in the cytoplasm, the fates of the mRNAs are also different. mRNA can be associated with ribosomes to make protein, stored in inclusion bodies for future usage, or targeted for degradation [28], which are also extensively regulated by RBPs. For example, protein translation is initiated via interactions of the 5′ 7-methylguanylate cap structure of the mRNA with the cap-binding protein eukaryotic initiation factor 4E (eIF4E) [29]. Interaction of eIF4E with the cap is the main control point in protein synthesis initiation. However, the speed of this process can be increased or stabilized by interaction of cap-binding protein complex with poly(A)-binding protein (PABP), a RBP that binds the poly(A) tail of the mRNA, to form a closed-loop structure between the 5′ and 3′ ends of the mRNA [30]. Further, mRNA stabilities in the cytoplasm are regulated by RBPs. Eukaryotic mRNAs are quite stable and often protected from exonuclease degradation, due to the presence of methylated guanosyl cap and the poly(A) tail, coated with PABP and other RBPs. Thus, degradation of the mRNA in cytoplasm often starts with removing the Cap and/or poly(A) tail in processes called de-capping and/or de-adenylation, followed by digestion through the action of exonucleases. However, the stabilities of a specific set of transcripts (like cytokines and chemokines involved in inflammation) are extensively regulated, because these mRNAs contain specific RNA sequences and/or structures, which are often located in the 3′ untranslated regions (3′ UTRs) [31]. These sequence motifs act as binding sites to two types of sequence-specific RBPs. The first type of RBP can recruit the RNA degradation machinery such as enzymes involved in de-capping, de-adenylation, and then exonucleases, to degrade mRNA. Alternatively, another type of RBP that lacks interaction with RNA degradation machinery can act to stabilize mRNA. For example, many cytokine and chemokine mRNAs contain AREs (AU-rich Elements) in their UTRs, making their stability highly regulated by RBPs [32]. ARE-binding proteins such as tristetraprolin (TTP), encoded by the Zfp36 gene of the Tis11-family, associate with these AREs and recruit the RNA degradation machinery, to degrade these mRNAs [33]. On the other hand, other ARE-binding proteins, such as HuR, stabilize their targets because they lack interactions with RNA degradation machinery [34].

RBPs also control the biogenesis of non-coding RNAs. MicroRNAs (miRNAs) are classes of small non-coding RNAs (~22 nucleotide RNA) that can negatively regulate mRNA stability and/or protein translation [35]. MiRNAs play important roles in cellular processes such as cell differentiation, proliferation, migration, apoptosis, and stress responses [36,37]. The miRNA biogenesis pathway is highly dependent on multiple dsRNA or ssRNA RBPs. miRNA production begins with the generation of the pri-miRNA transcript by RNA polymerase II/III, a dsRNA RBP. In the nucleus, an RNAase III enzyme, Drosha (also dsRNA RBP), and a co-factor protein, DiGeorge syndrome critical region gene 8 (DGCR8), convert the pri-miRNA to pre-miRNA through cleavage. Once pre-miRNA is formed, the molecule exits the nucleus into the cytoplasm via exportin 5, a RNA-binding protein. Another dsRNA RBP RNAase III, Dicer, coupled with Tar RNA-binding protein 2 (TARBP2), converts pre-miRNA to mature, stable miRNA through cleavage and splitting of phosphodiester bonds [38]. More importantly, RBPs can bind to the terminal loop or other elements within miRNA precursors that stimulate or block miRNA processing by different mechanisms at either the Drosha and/or Dicer level [39]. 

In addition to RBP function in affecting miRNA processing machinery, an emerging concept of miRNA–RBPs interactions and reciprocal regulations is recognized as an opportunity to better understand and potentially mitigate disease [40]. MiRNAs and RBPs are two major classes of regulatory molecules working post-transcriptionally on mRNAs. Based on the knowledge of how miRNA and RBPs regulate their target mRNAs, it is reasonable to predict that they may interact on the same mRNA and work together cooperatively or competitively to modulate mRNA stability and protein translation. Indeed, in cancer research, it is reported that RBPs can either enhance miRNA function by working as guides that mediate the opening of the mRNA structure [41,42] or counteract miRNA function by competitively binding to the potential binding sites (or the “seed” region) of miRNAs [43,44]. Reciprocal regulation between miRNAs and RBPs is well recognized, because miRNAs can regulate the expression of RBPs by translational repression, or in contrast, RBPs can regulate specific miRNA processing and, thus, expression [45,46]. In summary, a mature protein-coding mRNA may meet several regulators such as miRNAs and RBPs that will affect the production of the protein in a variety of ways. Further evaluation and a better understanding of the miRNA–RBPs interactions will lead to better understanding of the human disease.

Long non-coding RNA (lncRNAs) are >200 bp in length and are capable of regulating gene expression via diverse mechanisms at both transcriptional and post-transcriptional levels acting as signals, decoys, guides, and scaffolds [47,48]. They have been recognized as critical players in normal homeostasis, development, and diseases over the last decade. LncRNAs are transcribed by RNA Polymerase II and RNA Polymerase III from multiple loci of the genome. Overall, the generation of mature IncRNAs, similar to the generation of mRNA, involves extensive processing including 5′ cap, 3′ polyadenation, and splicing [49,50], in a RBP-dependent manner. Although mechanisms of biogenesis and regulation of different lncRNAs are still unclear, the next few years will deepen our insight into the role of RBPs in lncRNAs maturation and their function (also see above for the role of RBP and IncRNA in gene transcription) [51,52].

Circular RNA (circRNA) has been categorized as another type of noncoding RNA, featuring a covalently closed loop structure without a 5’ cap or 3’ poly(A) tail [53]. It has been demonstrated that circRNA expression is controlled by RBPs and RNA-editing enzymes, including QKI and adenosine deaminase acting on RNA (ADAR) [54,55]. Due to their recent discovery, the functions of circRNAs have not been fully elucidated. CircRNAs have been shown to act as miRNA sponges to regulate miRNA expression [56]. Other functions of circRNA, such as participating in the regulation of transcription and AS [57] and acting as protein-coding RNAs, have recently been described [58,59].

## 3. RBPS and Their Role in PH 

RBPs are increasingly recognized as important players in PH, as they play an important role in regulating the process of RNA metabolism and gene expression [60,61,62]. In the past decade, studies have elucidated the role of RBPs in PH through their actions in RNA metabolism, including transcription, RNA splicing, and stability (Table 1). New studies also support a concept wherein the function of RBP is controlled by RNA (Table 1, TLR3). We present our review of RBPs in PH in the following order: (1) control of gene transcription, (2) regulation of RNA splicing, (3) regulation of RNA stability in nucleus and cytoplasm, and (4) RNA acting on RBP to control RBP function. Finally, we present our recent observations of activated RBPs in the right ventricle (RV) from hypoxia-induced PH neonatal calves. 

### 3.1. The Role of RBPs in Regulating Gene Transcription in PH

Gene transcription regulation is traditionally known to be the function of DNA-binding TFs. Recent studies indicate that RNA-binding proteins are also involved in controlling gene transcription and chromatin structure via their interaction with RNA. The role of RBPs in controlling gene transcription in PH is only beginning to be investigated. It has been reported that reduced levels of the RBP splicing factor proline and glutamine-rich protein (SFPQ) promote activation of pulmonary artery adventitial fibroblasts via activating CD40 transcription in PH [63]. Activation of CD40 signaling in pulmonary artery adventitial fibroblasts resulted in an activated phenotype characterized by increased inflammation, proliferation, and migration [63]. SFPQ protein, also called polypyrimidine tract-binding protein-associated splicing factor (PSF), is a well-known RBP because SFPQ is required early in spliceosome formation and for splicing catalytic step II [73,74,75]. Overexpression of SFPQ suppressed CD40 transcription and reduced proliferation and migration of pulmonary adventitial fibroblasts. Using both ChIP and RNA immunoprecipitation assays, the authors concluded that SFPQ represses CD40 transcription in a RNA-dependent manner, via regulating histone acetylation on the CD40 promoter through cooperating with histone deacetylase 1 (HDAC1). This study demonstrated the role of RBP SFQP in regulating CD40 transcription and controlling the phenotype of pulmonary artery adventitial fibroblasts.

### 3.2. The Role of RBPs in Regulating RNA Splicing in PH

Multiple studies have established a critical role of splicing factors in regulating RNA splicing of mRNAs that are functionally important in PH pathophysiology. In this section, we will briefly summarize these studies in the order of RBPs in regulating RNA splicing as an AS repressor, an AS activator, and an auxiliary splicing protein that interacts with snRNPs to regulate snRNP’s association with 5′ and/or 3′ splice site(s).

#### 3.2.1. The RBP PTBP1 Regulates PKM Splicing by Acting as an AS Repressor in PH

It is established that hnRNPs act as AS repressors by binding to an intronic and/or exonic splicing silencer, to promote exon skipping. hnRNP-mediated AS is known to play an important role in human health and disease [76,77]. One splicing target of hnRNP in cancer cells is Pyruvate Kinase Muscle (PKM). PKM is an enzyme that plays a central role in controlling glycolysis. RNA splicing of PKM is important in controlling PKM activity. PKM can be spliced to produce the PKM1 (containing exon 9, but not exon 10) or PKM2 (containing exon 10, but not 9) isoforms [78,79], PKM1 and PKM2 are functionally different. In cancer cells, it is reported that heightened expression of PKM2 or an elevated PKM2/PKM1 ratio promotes glycolysis and is critical for the maintenance of cancer cell growth [80]. This mutually exclusive PKM splicing in cancer cells is controlled by three RBPs of the hnRNP family—PTBP1 (also known as hnRNPI), hnRNPA1, and hnRNPA2—that bind to splicing silencers flanking exon 9, preventing spliceosome recruitment to 5′ and 3′ splicing sites for exon 9. Thus, cancer cells with high levels of these hnRNP splicing proteins, express high levels of PKM2 isoform, resulting in a high PKM2/PKM1 ratio and high cancer cell proliferation [78,81,82,83].

The emerging “metabolic theory” of PH suggests that metabolic reprogramming underlies the pathology of this disease [84,85]. We and others have demonstrated that the persistently activated adventitial fibroblasts isolated from both human and bovine hypertensive pulmonary arterial walls (PH-Fibs) exhibit a hyperproliferative, apoptosis-resistant, pro-inflammatory phenotype and constitutive reprogramming of glycolytic and mitochondrial metabolism [86,87]. Furthermore, our group reported that PTBP1 controlled PKM mRNA AS thus playing a pivotal role in this metabolic shift. Specifically, our group reported an increased expression of PTBP1 and an elevated ratio of PKM2/PKM1 mRNA and protein levels in PH-Fibs [64,65]. Using a human PKM splicing reporter that contains exon 8 to exon 11 of the human PKM gene, we showed that PH-Fibs expressed much higher PKM2 transcripts, than that in the CO-Fibs. More importantly, PKM2 transcript levels in PH-Fibs were significantly reduced by PTBP1 silencing through siRNA technology, which supports the role of PTBP1 in regulating the PKM AS process in PH-Fibs. Finally, we found that siPTBP1 normalized the PKM2/PKM1 ratio and the glycolytic reprogramming and reduced cell proliferation in PH-Fibs (Figure 1A).

The critical role of PTBP1 as an AS regulator of PKM was also reported in PH blood outgrowth endothelial cells (BOECs) by another group [66]. They showed that heritable pulmonary arterial hypertension (HPAH) and idiopathic pulmonary artery hypertension (IPAH) BOECs exhibited increased expression of PTBP1 and PKM2, and a shift from oxidative phosphorylation to aerobic glycolysis. Knockdown of PTPB1 using siRNA corrected the metabolic imbalance and restored normal proliferation in HPAH BOECs (Figure 1A). 

Collectively, these studies support the significant fact that an increased PKM2/PKM1 ratio in PH vascular cells, similar to what is observed in cancer cells, leads to metabolic change, and ultimately cell proliferation. The studies provided strong evidence for aberrant expression of the RBP PTBP1, in controlling PKM splicing, which supports the idea of anti-RBP treatment being useful in PH (Figure 1A).

#### 3.2.2. The RBP SRSF2 Regulates BMPR2 Splicing by Acting as an AS Activator in PH 

At least 20 SR proteins have been identified from which a smaller group of 7 are termed “core” SR proteins including SRSF2 [26,88,89]. Both hnRNP and SR proteins are involved in AS; however, different from hnRNPs, SR proteins act as splicing activators by binding to splicing enhancers. Thus increased levels and/or activities of SR proteins typically increase the usage of specific exon. In contrast, reduced SR protein levels or activities often lead to skipping of exon in mature mRNA.

Genetic studies in familial pulmonary arterial hypertension (FPAH) have revealed heterozygous germline mutations (one mutated copy and one normal copy) in the bone morphogenetic protein type II receptor (BMPR2), a receptor for the transforming growth factor (TGF)-beta/bone morphogenetic protein (BMP) family. BMPR-mediated signaling plays a critical role in maintaining normal physiology of pulmonary vascular cells, as a significantly reduced BMPR2-mediated signal increases the susceptibility of pulmonary endothelial cells to apoptosis, but increases cell proliferation in pulmonary artery smooth muscle cells [90,91,92]. However, only 20% of individuals with BMPR2 heterozygous mutations develop symptoms of pulmonary arterial hypertension [93,94]. This incomplete penetrance has made genetic counseling difficult. Recent studies have uncovered that AS of the pre-mRNA from the normal BMPR2 allele may play a significant role in this incomplete penetrance.

In humans, BMPR2 has 13 exons and is alternatively spliced to produce two primary transcripts. Isoform-A contains all 13 exons, producing full length, functional BMPR2 protein. Isoform-B lacks exon 12 (Figure 1B). Studies have clearly shown that exon 12 is important for the proper function of BMPR2. Deletion of exon 12 is a common BMPR2 mutation found in HPAH patients. More importantly, this B isoform not only lacks proper function but can also disrupt the function of BMPR2 A isoform in a dominant-negative fashion. Cogan and colleagues reported that affected BMPR2 mutation carriers (who develop the clinical disease) have higher amounts of isoform-B mRNA relative to isoform-A compared to unaffected BMPR2 mutation carriers [67]. 

They found that BMPR2 exon 12 contains a splice enhancer that is the binding site for splicing activator SRSF2, and SRSF2 expression is significantly reduced in affected BMPR2 mutation carriers. Indeed, they found that knocking down SRSF2 in pulmonary microvascular endothelial cells resulted in elevated levels of isoform-B compared to isoform-A. This observation indicated that the increased isoform-B formation (skipping of exon 12) from the normal copy of BMPR2 DNA, and thus almost complete loss of BMPR2 signaling in these affected patients with BMPR2 mutation, is due to reduced expression of SRSF2. Future studies are needed to address the molecular mechanism underlying the reduced expression of SRSF2.

In conclusion, Cogan and colleagues showed that AS of BMPR2 has a significant role in HPAH penetrance of BMRP2 heterozygous germline mutation carriers. In mutation carriers with reduced SRSF2 expression, the BMPR2 pre-mRNA from the normal BMRP2 allele will splice in a manner that results in more isoform-B transcript relative to isoform-A. These mutation carriers are more likely to develop PAH (Figure 1B). Thus, checking SRSF2 protein expression may provide BMRP2 heterozygous germline mutation carriers more information during genetic counseling.

#### 3.2.3. The RBPs RBM25 and LUC7L3 Regulate SCN5a Splicing as Splicing Auxiliary Proteins of the Spliceosome in Heart Failure and PH 

Sodium voltage-gated channel alpha subunit 5 (SCN5a) is the cardiac sodium channel (Nav 1.5), a protein responsible for generating the main current for excitation propagation in cardiomyocytes. The critical role of SCN5a is indicated by the fact that decreased SCN5a levels are associated with human heart failure (HF) and the increased risk of sudden death in HF [95,96]. However, the generation of non-functional SCN5a splice variants (SV), rather than reduced gene transcription, is often associated with human HF [95,96,97]. It is reported that increased levels of splicing factors RNA-binding motif protein 25 (RBM25) and LUC7-like three pre-mRNA splicing factor (LUC7L3), in response to hypoxia and/or increased angiotensin II-mediated signal, are responsible for increased levels of SCN5a SV. Both RBM25 and LUC7L3 belong to a family of RBPs functioning in pre-mRNA splicing and are associated with multiple splicing components such as U1 snRNP. It has been demonstrated that SCN5a SV result from splicing using cryptic splice sequences in the terminal exon of SCN5A (exon 28), due to increased levels of RBM25 and LUC7L3 and increased usage of the cryptic 3′ splicing site in the terminal exon [95].

In a recent pilot study of eleven Group 1 PAH patients and eleven age and sex-matched controls, Banerjee and colleagues reported that circulating SCN5a SV mRNA levels were much higher in PAH patients, compared with controls. Further, they observed significant inverse correlations between SCN5a SV expression levels and baseline mPAP and PVR [68]. Thus, although the splicing factors regulating SCN5a SV in PH are still unknown, it supports the idea that interruption of this abnormal mRNA processing may reduce the mortality of PH since the reduction in channels is associated with sudden death in PH patients.

### 3.3. The Role of RBPs in Regulating RNA Stability in PH

#### 3.3.1. The RBP ZFC3H1 Regulates Nuclear RNA Stability in PASMCs 

Increased proliferation of pulmonary artery smooth muscle cells (PASMCs) plays an important role in PH vascular remodeling. To discover a novel drug for PAH that inhibits PASMC proliferation, Kurosawa, and colleagues performed high-throughput screening of 5562 compounds in PASMCs from patients with PAH (PAH-PASMCs) [69]. Celastramycin was found to be a compound that could significantly inhibit PAH-PASMC proliferation, by reducing cytosolic reactive oxygen species (ROS), and reducing RNA levels of multiple TFs, such as HIF1a, NF-KB, and bromodomain-containing protein 4 (BRD4), that are known to regulate smooth muscle cell proliferation and ROS generation. Furthermore, they reported that Celastramycin’s anti-PH effects on PAH-PASMCs were mediated by the RBP zinc finger C3H1 domain-containing protein (ZFC3H1). ZFC3H1 is an adaptor protein that mediates the interaction between nuclear RNAs and the exonucleases complex called exosome in the nucleus and plays a crucial role in the degradation of nuclear RNAs [98,99,100]. In this study, the authors concluded that ZFC3H1, as a binding partner of Celastramycin, mediated the inhibition of HIF1a and BRD4 RNAs through regulating nuclear RNA stability.

#### 3.3.2. The RBP HuR Regulates Cytoplasm sGC-a1 mRNA in Pulmonary Arteries 

HuR is a member of the mammalian Hu/embryonic lethal abnormal vision (ELAV) protein family, originally identified as antigens associated with paraneoplastic neurological syndrome [101]. HuR is mainly located in the cytoplasm. Binding of HuR to ARE-containing mRNAs in cytoplasm prevents RNA degradation and increases mRNA stability. It was reported that HuR regulates soluble guanylyl cyclase α1 (sGC-α1) levels in a post-transcriptional manner in rat models of genetic hypertension [102,103]. sGC is a receptor for nitric oxide (NO). The NO signal transduction pathway promotes smooth muscle (SM) relaxation. In addition to its vasodilator function, NO/sGC/PKG signaling also was reported to antagonize the proliferative responses in vascular smooth muscle cells [104,105]. Thus, in PH, NO/sGC signaling is decreased, and many current therapies for PH (sildenafil, tadalafil, and riociguat) are directed at increasing NO/sGC signaling [106,107]. Excitingly, de Frutos and colleagues [70] demonstrated that short-term (2 days) exposure of mice to hypoxia increased nuclear translocation of HuR protein in pulmonary arteries resulting in reduced HuR in the cytoplasm, decreased sGC-α1 mRNA levels, and attenuated NO/sGC signaling. This finding suggests that HuR might be an additional therapeutic target as a sGC stimulator, especially for acute PH.

### 3.4. RBP TLR3 Stimulation by dsRNA Protects against PH in Rat Models: A New Concept Wherein the Function of RBP Is Controlled by RNA

Detection of foreign nucleic acids, such as dsRNA, that often accumulate during viral infection, is a central mechanism for innate immune defense in most organisms. Accumulating evidence suggests that the innate immune system can be stimulated not only by pathogen-derived dsRNA but also by host-generated dsRNAs derived from cellular mRNA [108] and RNA released from dying cells during tissue necrosis [109]. Toll-like receptor 3 (TLR3), also known as CD283, is a member of the toll-like receptor family of pattern recognition receptors in the innate immune system [15,110,111]. However, unlike most other TLRs that sense dsRNA and promote inflammation, it is reported that TLR3 is required for protective immunity against virus infection [112,113] and has been implicated in vascular protection [114].

Several studies have demonstrated that vascular remodeling in PH is characterized by the accumulation of perivascular inflammatory cells in animal models of PH, with similar perivascular inflammatory infiltrates observed in human PAH [2,115,116]. However, targeting inflammation-related TFs such as NFKB or Stat3 has not been successfully implemented as a therapeutic approach. Farkas and colleagues [71] demonstrated that TLR3 expression is significantly decreased in pulmonary artery endothelial cells (PAECs), remodeled pulmonary arteries, and lung tissue of patients with PH and show that TLR3 deficiency increases susceptibility to apoptosis and pulmonary hypertension in vitro and in vivo. Consistent with a protective role of TLR3 against PH development, TLR3^−^/^−^ mice exhibited more severe pulmonary hypertension upon exposure to chronic hypoxia/SU5416 while TLR3 activation using a high-dose synthetic ligand of dsRNA, polyinosinic:polycytidylic acid (poly[I:C]), reduced established experimental PH in two rat models [71,72]. Further, in vitro and in vivo experiments identified that the protective activity of TLR3 stimulation was mediated by promoting IL-10 expression and thus IL-10-mediated inhibition of inflammation (control of CXCL10 [IP-10] expression) [71,72]. In summary, TLR3, as an anti-inflammatory RBP, could be a therapeutic target in PH.

### 3.5. Dysregulated RBPs at the mRNA Level Are Observed in RV in Hypoxia-Induced PH Calf 

As mentioned above, the recent study by de Bruin and colleagues provided the profile of heart-associated RBPs in cardiomyocytes. It defined the important role of RBPs in heart development, and also demonstrated how altered RBP expression could impact cardiac function [19]. Consistent with their observations, our most recent cDNA microarray data demonstrated multiple differentially expressed heart-associated RBPs in RV using our established model of hypoxia-induced PH neonatal calves (*n* = 5) compared to the healthy controls (*n* = 5). The changes of heart-associated RBPs in PH calf RV include the following: (1) increased levels of QKI, which has been demonstrated to regulate cardiomyocyte-mediated remodeling of the heart following injury by regulating the AS of myocardin pre-mRNA [19]. (2) Increased muscleblind-like splicing regulator 2 (MBNL2). Re-expression of embryonic RNA isoform of MBNL2 has been reported in rat models of cardiac hypertrophy [117]. (3) Increased CUGBP Elav-like family member 1 (CELF1). CELF1 has been reported to regulate mRNA stability and translation of target genes [118]. The upregulated CELF1 in adult cardiomyocytes led to arrhythmia, dilated cardiomyopathy, and eventual heart failure [119,120]. Additionally, mouse models have shown that CELF1 overexpression in the heart results in splicing defects, which leads to cardiomyopathy and ultimately cardiac failure [118,120]. (4) Dysregulated RNA-binding motif (RBM) family including RBM5, RBM18, RBM25, RBM3, RBM12, RBM22, and RBM23, which are well-characterized splicing regulators in the heart and play an important role in cardiac disease [121]. These results suggest that RBPs are activated in the RV at an early stage of PH and may contribute to compensatory RV adaptation in response to hemodynamic pressure overload. More study is necessary to evaluate the role of RBPs in adaptive versus maladaptive RV remodeling during the progression of PH. 

## 4. Disruption of Other Aspects of RNA Metabolism in PH

Numerous studies have demonstrated that many miRNAs are dysregulated in patients with PAH and experimental PH. Changes in miRNA expression have been observed in all pulmonary vascular cell types and shown to contribute to many aspects of the dysregulated phenotypes observed in PH [64,122,123]. It is reported that RBPs are necessary for the production of miRNAs and miRNA-mediated gene expression in mammalian cells [11,124,125]. There is no direct evidence of RBP function in miRNA maturation and function in specific PH research. It has been shown that Dicer is downregulated by hypoxia in human umbilical vein endothelial cells (HUVECs), leading to a subsequent decrease in the levels of various mature miRNAs [126]. Further research should explore the role of miRNA-associated RBPs in PH. 

There is increasing evidence supporting a key role of lncRNAs in pathogenesis of PAH [127]. A number of lncRNAs such as H19 [128], MEG3 [129], and UCA1 [130] have been shown to regulate PASMC proliferation, migration, and apoptosis. LncRNA MALAT1 [131,132] and GATA6-AS [133] induce mesenchymal transition (EndMT) in pulmonary artery endothelial cells, and this could contribute to the pathology of PAH. These findings support the role of lncRNAs as key participants in the regulation of major cellular phenotypes involved in the pathogenesis of PAH. Further investigation to determine the role of RBPs in regulating the expression/activity of lncRNA may provide new insight into PH pathogenesis. 

Limited studies have been conducted regarding the role of circRNAs in PH development. Recently, to determine the expression profile of circRNA in PH lung, Wang, and colleagues [134] performed microarray analysis in the lung tissues from mice with hypoxia-induced PH. They found 23 significantly upregulated and 41 significantly downregulated circRNAs and further predicted circRNA–miRNA interactions by putative miRNAs binding sites sequencing. Another group found that the expression of hsa_circ_0016070 was increased in COPD patients with PH compared to COPD patients without PH. Furthermore, they demonstrated that hsa_circ_0016070 enhances vascular remodeling in PAH by promoting PASMC proliferation through the miR-942/CCND1 pathway. In summary, circRNAs are involved in the pathogenesis of PH, at least partly through circRNA–miRNA interactions. Investigation focusing on the role of RBPs in controlling non-coding RNA expression and interactions is needed.

## 5. Summary and Future Perspectives 

Emerging in vitro and in vivo experimental data indicate that RBPs are dysregulated in PH and have significant effects on pulmonary vascular cell phenotype and overall disease pathophysiology. RNA AS, transcription, and RNA stability defects are the major types of RBP–RNA interaction events contributing to PH development based on observations to date. This highlights the possibility of targeting RBPs to restore normal pulmonary vascular phenotypes. For instance, RNA interference-based approaches have been used in in vitro experiments. Knockdown of PTBP1 [64,65,66] and ZFC3H1 [69] using specific siRNA was sufficient to normalize the hypertensive phenotypes of pulmonary artery vascular cells, including hyperproliferation, apoptosis resistance, increased production of proinflammatory proteins and metabolic reprogramming. 

On the other hand, Lentivirus-mediated SFPQ overexpression [63] and stimulation of TLR3 using a synthetic ligand of dsRNA poly[I:C] [71,72] also attenuated PH characteristics in pulmonary artery vascular cells in vitro as well as in PH animal models. Importantly, these findings, when considered in the context of the recent development of sophisticated delivery methods for RNA-based therapeutics [135,136,137], provide the interesting possibility of new treatment options for PH by targeting RBP expression or activity or targeting specific RBP-mediated events. Further studies are needed to elucidate accurate molecular mechanisms of RBP–RNA interaction in PH because most RBPs are involved in multiple steps of RNA processing and interact with a large number of RNA targets. 

## Figures and Tables

**Figure 1 ijms-21-03757-f001:**
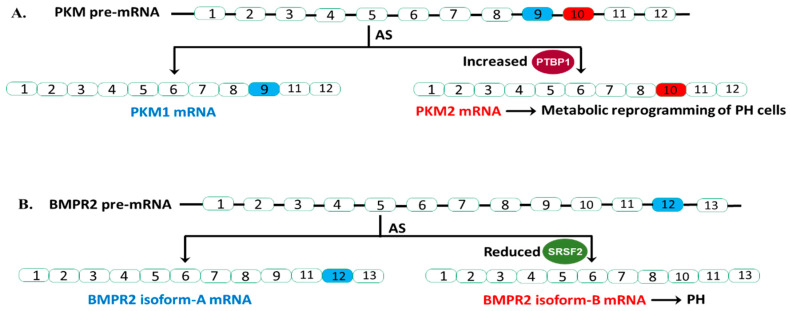
AS events in PKM and BMPR2 and their relevance in PH. (**A**). Increased PTBP1 (splicing repressor) inhibit the inclusion of exon 9 of PKM pre-mRNA, resulting in increased generation of PKM2 isoform and alteration of phenotypes of PH cells. (**B**). Reduced SRSF2 (splicing activator) levels in a subset of BMPR2 heterozygous mutation carriers increase BMPR2 isoform-B generation, lead to almost complete loss of BMRP2 function (one copy is mutated, another copy generated the B isoform) and PH.

**Table 1 ijms-21-03757-t001:** Role of RBPS in pulmonary hypertension.

RBP	Experimental System	Target mRNA(s)	RNA Metabolism Alteration	Phenotype/Function	Reference
SFPQ	Pulmonary artery adventitial fibroblasts isolated from rat	CD40	Transcription	Reduced levels of RBP SFPQ promote activation of pulmonary artery adventitial fibroblasts via activating CD40 transcription	[63]
PTBP1	Human patient samples; human and bovine pulmonary artery fibroblasts; human PAECs; human BOECs	PKM (PKM1 and PKM2)	Splicing	Increased levels of splicing repressor PTBP1 inhibit the usage of exon 9 of PKM pre-mRNA, resulting in increased generation of PKM2 and alteration of phenotypes of PH vascular and circulating cells.	[64,65,66]
SRSF2	Lymphocytes (CLs) from BMPR2 mutation-positive HPAH patients and unaffected carriers; human pulmonary microvascular endothelial cells (PMVECs)	BMPR2 (isoform B and A)	Splicing	Reduced levels of splicing activator SRSF2 increase the levels of non-functional BMPR2 B isoform in PH cells from affected BMPR2 mutation carriers (who develop PH), providing an explanation of the reduced penetrance among BMPR2 heterozygous mutation carriers.	[67]
HF related splicing factor(s)	Peripheral blood mononuclear cells (PBMCs) isolated from Patients with Group 1 PAH and Controls	SCN5a	Splicing	Increased levels of RBP (might be RBM25 and LUC7L3) promote the generation of non-functional splicing variant of SCN5a in heart failure and PH	[68]
ZFC3H1	PASMCs from patients with PAH	BRD4 and HIF1α	Stability	As a binding partner of Celastramycin, ZFC3H1 mediates inhibition of BRD4 and HIF-1a by Celastramycin treatment through regulating the degradation of nuclear RNAs in PASMCs	[69]
HuR	Pulmonary arteries of hypoxic mice PH model	sGC-α1	Stability	Increased translocation of HuR protein from cytoplasm to nucleus de-stabilizes sGC-a1 mRNA in mouse pulmonary arteries, thus reducing NO/sGC signaling in response to short-term hypoxia	[70]
TLR3	PAECs, Pulmonary arteries and lung tissue of patients with PH; rat endothelial cells; Chronic hypoxia and SU5416 TLR3^−^/^−^ and TLR3^+^/^+^ mice; chronic hypoxia/SU5416 rats	IL-10 CXCL10 (IP10)	Transcription	TLR3 expression is significantly reduced in PAECs, remodeled arteries, and lung tissue of patients with PH. TLR3 activation by a synthetic ligand of double-stranded RNA (poly[I:C]), ameliorated established experimental PH in rat models.	[71,72]

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
