# Peer review of "RNA-Binding Proteins in Pulmonary Hypertension"

_ijms, 2020, doi:10.3390/ijms21113757_

Round 1

Reviewer 1 Report

The review is overall well-written and easy to follow. It provides interesting insight into a less researched, but promising field of RNA binding proteins and their function in pulmonary hypertension.

The structure of the paper is logical and gives clear summary of the available knowledge, drawing attention to open questions and possible new therapeutic targets.

My suggestion is that the authors include a separate section in the introduction where they give a short overview of the pathophysiology of PH. Given the length and detail the RNA regulation is discussed, a part where the main topic of the paper is introduced is missing and would improve the quality of the manuscript.

Other than this, I have only some minor comments:

  1. Since RNA splicing is discussed in detail in the introduction, the lines 223-225 are redundant and can be omitted.
  2. Not all abbreviations are explained.

  3. There are some typing errors and incorrect sentences, so a spell check is recommended. Some examples, but there might be others I didn't notice:

Line 52: identify (should be identitiy)

Line 273: The RBP SRSF2 regulates BMPR2 splicing by as an AS activator in PH ('by' is unnecessary or something is missing)

Line 317: ... of a non-functional SCN5a splice variants (should be variant)

Author Response

The review is overall well-written and easy to follow. It provides interesting insight into a less researched, but promising field of RNA binding proteins and their function in pulmonary hypertension.

The structure of the paper is logical and gives clear summary of the available knowledge, drawing attention to open questions and possible new therapeutic targets.

Response: Thank you for your time and effort in reviewing our manuscript and really appreciate the positive comments on the manuscript.

My suggestion is that the authors include a separate section in the introduction where they give a short overview of the pathophysiology of PH. Given the length and detail the RNA regulation is discussed, a part where the main topic of the paper is introduced is missing and would improve the quality of the manuscript.

Response: We totally agree with your suggestion and now include a brief introduction regarding the pathophysiology of PH in first paragraph (line 48-53) in the revised manuscript.

Minor comments:

  1. Since RNA splicing is discussed in detail in the introduction, the lines 223-225 are redundant and can be omitted.

Response: We agree and deleted the sentences of “Splicing or alternative splicing is a complex process that is regulated by RNA sequence (5’ splicing site, 3’ splicing site, and exon and intron splicing enhancers and silencers) and the proteins (components of spliceosome and AS activators and repressors) that recognize and bind to these RNA sequences”

  1. Not all abbreviations are explained.

Response: We improved all abbreviations and make them consistent throughout the manuscript.

  1. There are some typing errors and incorrect sentences, so a spell check is recommended.

Response: We agree and performed a spelling check of the entire manuscript and corrected the errors.

Line 52: identify (should be identity)

Response: we have changed “identify” to “Identity”

Line 273: The RBP SRSF2 regulates BMPR2 splicing by as an AS activator in PH ('by' is unnecessary or something is missing)

Response: We have changed the sentence to “:The RBP SRSF2 regulates BMPR2 splicing by acting as an AS activator in PH” 

Line 317: ... of a non-functional SCN5a splice variants (should be variant)

Response: we have deleted “a”

Reviewer 2 Report

In this review manuscript, Zhang et al describe the role of post-transcriptional regulatory mechanism in the pathogenesis of PH. The authors focus on various RBPs and comprehensively present the importance of these RBPs in PH-relevant pathogenesis. 

The review is well written and important cases were well presented. The manuscript will be presented better with more figures. Particularly, the authors describe complicated AS events in PKM and other genes and their relevance in PH. Figures will help the audience understand better.

Also, there are numerous mistakes across the manuscript and I put some examples as minor points below. The authors need to check them out. 

Minor points.

-In line 80, lifecycle should be life cycle.

-In line 87, please define ChIP-seq

-The sentence starting in line 113 and ends in line 115 is not accurate. Please remove -noncoding in the sentence.

-In line 109, alternative splicing is defined to AS. However, the rest of the manuscript, AS and alternative splicing were mixed up. Please be consistent.

-In line 117, hnRNPS should be hnRNPs.

-In line 190, please define ADAR.

-In line 209 and 214, SFPQ should be defined in line 209.

-In line 220, please define HDAC1.

-The sentences in line 230-235 are redundant as a similar content was described earlier in the manuscript.

-In line 260, a typo for PKM1. Currently, it is PM1.

-In line 262-263, please define PH BOEC and HPAH.

-In line 276, the authors used traditional nomenclature of splicing factors. Just like they have used SRSF2 as an example, I recommend to use both official and traditional names of these splicing factors.

-The sentence in line 277-278 is redundant as a very similar sentence has appeared earlier.

-In line 319, please define RBM and LUC7L3.

-In line 339 and 341, define ROS.

-In line 341, please define SMC.

-In line 348, HuR was defined earlier so it does not need to redefine here.

-In line 398, Quaking was defined here but Quaking also appears earlier in the manuscript.

Author Response

In this review manuscript, Zhang et al describe the role of post-transcriptional regulatory mechanism in the pathogenesis of PH. The authors focus on various RBPs and comprehensively present the importance of these RBPs in PH-relevant pathogenesis. 

Response: Thank you for your time and effort in reviewing our manuscript. We appreciate the positive comments regarding the manuscript.

The review is well written and important cases were well presented. The manuscript will be presented better with more figures. Particularly, the authors describe complicated AS events in PKM and other genes and their relevance in PH. Figures will help the audience understand better.

Response: We thank you for your excellent suggestion and now include a new Fig (Fig 1) to illustrate the AS events in PKM and BMPR2 genes and their relevance in PH. We cite the new figure and relevant work in the revised manuscript in line 274 and line 310. 

Also, there are numerous mistakes across the manuscript and I put some examples as minor points below. The authors need to check them out. 

Response: Thank you very much for pointing out these mistakes. We corrected the mistakes you listed and performed a spelling check of the whole manuscript and corrected further errors. We improved all abbreviations and made them consistent throughout the manuscript. We removed the redundant sentences as suggested.

-In line 80, lifecycle should be life cycle.

Response: “Lifecycle” has been changed to “life cycle

-In line 87, please define ChIP-seq

Response: defined

-The sentence starting in line 113 and ends in line 115 is not accurate. Please remove -noncoding in the sentence.

Response: We have modified the sentences to “ AS allows each gene to encode multiple mRNA variants that may code no proteins or proteins with different or even opposite activities.”

-In line 109, alternative splicing is defined to AS. However, the rest of the manuscript, AS and alternative splicing were mixed up. Please be consistent.

Response: we used AS after AS was defined. Thank you.

-In line 117, hnRNPS should be hnRNPs.

Response: we have changed “hnRNPS” to “hnRNPs”

-In line 190, please define ADAR.

Response: defined

-In line 209 and 214, SFPQ should be defined in line 209.

Response: defined

-In line 220, please define HDAC1.

Response: defined

-The sentences in line 230-235 are redundant as a similar content was described earlier in the manuscript.

Response: we have deleted the sentences of “hnRNPs represent a large family of RBPs that contribute to multiple aspects of nucleic acid metabolism including alternative splicing, mRNA stabilization, and transcriptional and translational regulation”

-In line 260, a typo for PKM1. Currently, it is PM1.

Response: We have changed “PM1” to “PKM1”

-In line 262-263, please define PH BOEC and HPAH.

Response: defined

-In line 276, the authors used traditional nomenclature of splicing factors. Just like they have used SRSF2 as an example, I recommend to use both official and traditional names of these splicing factors.

Response: The sentence has been changed to “At least 20 SR proteins have been identified from which a smaller group of 7 are termed “core” SR proteins including SRSF2” in the revised version, because most SR proteins were not further described.

-The sentence in line 277-278 is redundant as a very similar sentence has appeared earlier.

Response: we have deleted the sentence of “Serine-arginine rich proteins (SR protein) are a conserved family of proteins involved in RNA splicing and associated with many human diseases” in the revised version.

-In line 319, please define RBM and LUC7L3.

Response: defined

-In line 339 and 341, define ROS.

Response: defined

-In line 341, please define SMC.

Response: defined

-In line 348, HuR was defined earlier so it does not need to redefine here.

Response: we have used “HuR” in the revision.

-In line 398, Quaking was defined here but Quaking also appears earlier in the manuscript.

Response: We have defined Quaking as QKI when it first appeared, then used QKI in later paragraphs.